# MULTI-OBJECTIVE OPTIMIZATION VIA EQUIVARIANT DEEP HYPERVOLUME APPROXIMATION

**Jim Boelrijk**
AI4Science Lab, AMLab
Informatics Institute, HIMS
University of Amsterdam
j.h.m.boelrijk@uva.nl

**Bernd Ensing**
AI4Science Lab
HIMS
University of Amsterdam
b.ensing@uva.nl

**Patrick Forré**
AI4Science Lab, AMLab
Informatics Institute
University of Amsterdam
p.d.forre@uva.nl

## ABSTRACT

Optimizing multiple competing objectives is a common problem across science and industry. The inherent inextricable trade-off between those objectives leads one to the task of exploring their Pareto front. A meaningful quantity for the purpose of the latter is the hypervolume indicator, which is used in Bayesian Optimization (BO) and Evolutionary Algorithms (EAs). However, the computational complexity for the calculation of the hypervolume scales unfavorably with increasing number of objectives and data points, which restricts its use in those common multi-objective optimization frameworks. To overcome these restrictions, previous work has focused on approximating the hypervolume using deep learning. In this work, we propose a novel deep learning architecture to approximate the hypervolume function, which we call DeepHV. For better sample efficiency and generalization, we exploit the fact that the hypervolume is scale equivariant in each of the objectives as well as permutation invariant w.r.t. both the objectives and the samples, by using a deep neural network that is equivariant w.r.t. the combined group of scalings and permutations. We show through an ablation study that including these symmetries leads to significantly improved model accuracy. We evaluate our method against exact, and approximate hypervolume methods in terms of accuracy, computation time, and generalization. We also apply and compare our methods to state-of-the-art multi-objective BO methods and EAs on a range of synthetic and real-world benchmark test cases. The results show that our methods are promising for such multi-objective optimization tasks.

## 1 INTRODUCTION

Imagine, while listening to a lecture you also quickly want to check out the latest news on your phone, so you can appear informed during lunch. As an experienced listener, who knows what lecture material is important, and an excellent reader, who knows how to scan over the headlines, you are confident in your abilities in each of those tasks. So you continue listening to the lecture, while scrolling through news. Suddenly you realize you need to split focus. You face the unavoidable trade-off between properly listening to the lecture while slowly reading the news, or missing important lecture material while fully processing the news. Since this is not your first rodeo, you learned over time how to transition between these competing objectives while still being optimal under those trade-off constraints. Since you don't want to stop listening to the lecture, you decide to listen as closely as possible, while still making some progress in reading the news. Later, during lunch, you propose an AI that can read the news while listening to a lecture. The question remains of how to train an AI to learn to excel in different, possibly competing, tasks or objectives and make deliberate well-calibrated trade-offs between them, whenever necessary.

Simultaneous optimization of multiple, possibly competing, objectives is not just a challenge in our daily routines, it also finds widespread application in many fields of science. For instance, in machine learning (Wu et al., 2019; Snoek et al., 2012), engineering (Liao et al., 2007; Oyama et al., 2018), and chemistry (O'Hagan et al., 2005; Koledina et al., 2019; MacLeod et al., 2022; Boelrijk et al., 2021; 2023; Buglioni et al., 2022).

The challenges involved in this setting are different from the ones in the *single-objective* case. If we are only confronted with a single objective function $f$ the task is to find a point $\mathbf{x}^*$ that maximizes $f$: $\mathbf{x}^* \in \arg\max_{\mathbf{x} \in \mathcal{X}} f(\mathbf{x})$, only by iteratively proposing input points $\mathbf{x}_n$ and evaluating $f$ on $\mathbf{x}_n$ and observing the output values $f(\mathbf{x}_n)$. Since the usual input space $\mathcal{X} = \mathbb{R}^D$ is uncountably infinite we can never be certain if the finite number of points $\mathbf{x}_1, \ldots, \mathbf{x}_N$ contain a global maximizer $\mathbf{x}^*$ of $f$. This is an instance of the *exploration-vs-exploitation* trade-off inherent to Bayesian optimization (BO) (Snoek et al., 2012), active learning, (Burr, 2009) and reinforcement learning(Kaelbling et al., 1996).

In the *multi-objective* (MO) setting, where we have $M \geq 2$ objective functions, $f_1, \ldots, f_M$, it is desirable, but usually not possible to find a single point $\mathbf{x}^*$ that maximizes all objectives at the same time:

$$\mathbf{x}^* \in \bigcap_{m=1}^{M} \underset{\mathbf{x} \in \mathcal{X}}{\arg\max} \, f_m(\mathbf{x}). \tag{1}$$

A maximizer $\mathbf{x}_1^*$ of $f_1$ might lead to a non-optimal value of $f_2$, etc. So the best we can do in this setting is to find the set of *Pareto points* of $F = (f_1, \ldots, f_M)$, i.e. those points $\mathbf{x} \in \mathcal{X}$ that cannot be improved in any of the objectives $f_m$, $m = 1, \ldots, M$, while not lowering the other values:

$$\mathcal{X}^* := \{\mathbf{x} \in \mathcal{X} \mid \nexists \mathbf{x}' \in \mathcal{X}. \, F(\mathbf{x}) \prec F(\mathbf{x}')\}, \tag{2}$$

Figure 1: Illustration of Pareto front and Hypervolume.

where $\mathbf{y} \preceq \mathbf{y}'$ means that $y_m \leq y'_m$ for all $m = 1, \ldots, M$ and $\mathbf{y} \prec \mathbf{y}'$ that $\mathbf{y} \preceq \mathbf{y}'$, but $\mathbf{y} \neq \mathbf{y}'$. In conclusion, in the *multi-objective* setting we are rather concerned with the *exploration of the Pareto front*:

$$\mathcal{P}^* := F(\mathcal{X}^*) \subseteq \mathbb{R}^M =: \mathcal{Y}, \tag{3}$$

which often is a $(M-1)$-dimensional subspace of $\mathcal{Y} = \mathbb{R}^M$. Success in this setting is measured by how "close" the *empirical Pareto front*, based on the previously chosen points $\mathbf{x}_1, \ldots, \mathbf{x}_N$:

$$\hat{\mathcal{P}}_N := \{F(\mathbf{x}_n) \mid n \in [N], \, \nexists n' \in [N]. \, F(\mathbf{x}_n) \prec F(\mathbf{x}_{n'})\} \subseteq \mathcal{Y}, \tag{4}$$

is to the 'true' Pareto front $\mathcal{P}^*$, where $[N] := \{1, \ldots, N\}$. This is illustrated in Fig. 1, where the blue points form the empirical Pareto front and where the black line depicts the true Pareto front.

Since the values $F(\mathbf{x}_n)$ can never exceed the values of $\mathcal{P}^*$ w.r.t. $\prec$ one way to quantify the closeness of $\hat{\mathcal{P}}_N$ to $\mathcal{P}^*$ is by measuring its *hypervolume* $\mathrm{HV}_{\mathbf{r}}^M(\hat{\mathcal{P}}_N)$ inside $\mathcal{Y}$ w.r.t. a jointly dominated reference point $\mathbf{r} \in \mathcal{Y}$. This suggests the multi-objective optimization strategy of picking the next points $\mathbf{x}_{N+1} \in \mathcal{X}$ in such a way that it would lead to a maximal improvement of the previously measured hypervolume $\mathrm{HV}_{\mathbf{r}}^M(\hat{\mathcal{P}}_N)$. This is illustrated in Fig. 1, where the hypervolume (i.e., the area in 2D) is shown for the empirical Pareto front (blue dots). Adding an additional point to the empirical Pareto front ($y_8$ in Fig. 1), increases the hypervolume indicated by the green area.

Unfortunately, known algorithms for computing the hypervolume scale unfavorably with the number of objective functions $M$ and the number of data points $N$. Nonetheless, finding a fast and scalable computational method that approximates the hypervolume reliably would have far-reaching consequences, and is an active field of research. Computation of the hypervolume has complexity of $O(2^N NM)$ when computed in a naive way, however more efficient exact algorithms have been proposed such as WFG ($O(N^{M/2} \log N)$,While et al. (2012)) and HBDA ($O(N^{M/2})$, Lacour et al. (2017)). However, these computational complexities are still deemed impractical for application in EAs (Tang et al., 2020), where computational overhead typically is required to be low. Also in BO, where one is typically less concerned with the computational overhead, faster hypervolume methods would be beneficial. For instance, the state-of-the-art expected hypervolume improvement (qEHVI) (Daulton et al., 2020) is dependent on many hypervolume computations, greatly restricting its use to the setting of low $M$ and $N$. In addition, the authors of the recently proposed MORBO (Daulton et al., 2022), which is current state-of-the-art in terms of sample-efficiency and scalability to high $M$ and $N$, identified the computational complexity of the hypervolume as a limitation.

Therefore, efforts have been made to approximate the hypervolume. The method FPRAS (Bringmann & Friedrich, 2010), provides an efficient Monte Carlo (MC) based method to approximate the hypervolume, of complexity $O(NM/\epsilon)$, with the error of $\pm\sqrt{\epsilon}$. In addition, at time of this work, a hypervolume approximation based on DeepSets (Zaheer et al., 2017) was proposed called HV-Net (Shang et al., 2022a). HV-Net uses a deep neural network with permutation invariance (i.e.

DeepSets) w.r.t. to the order of the elements in the solution set to approximate the hypervolume of a non-dominated solution set. It showed favorable results on both the approximation error and the runtime on MC and line-based hypervolume approximation methods.

Other methods try to avoid hypervolume computation entirely and use other metrics to guide optimization (Hernandez-Lobato et al., 2016). Such as ParEgo (Knowles, 2006), which randomly scalarizes the objectives to generate diverse solutions. Other approaches focus on distance metrics regarding the empirical Pareto front (Rahat et al., 2017; Deb et al., 2002a). However, methods using the hypervolume to guide optimization, are typically superior. This is shown both in the setting of EAs and for BO. For example, a plethora of work shows that SMS-EMOA (Beume et al., 2007) is more sample-efficient compared to MO EAs using alternative infill criterion (Narukawa & Rodemann, 2012; Tang et al., 2020). Also, qEHVI and MORBO have strong empirical performance compared to alternatives (Daulton et al., 2020; 2022).

In this paper, we develop DeepHV, a novel hypervolume approximation approach based on deep learning. As an input the model requires a non-dominated solution set after which the model outputs the predicted hypervolume of this solution set. By using specialized layers, DeepHV incorporates mathematical properties of the hypervolume, such as scaling equivariance of the hypervolume w.r.t. the scaling of the objectives in the solution set. In addition, the DeepHV is permutation invariant to the order of the elements and objectives in the solution set. Thus leveraging additional symmetry properties of the hypervolume compared to HV-Net. We show that our method allows for development of a single model that can be used for multiple objective cases up to $M = 10$. We obtain improved performance compared to HV-Net on unseen test data. Moreover, we perform an ablation study to show that the additional symmetry properties improve model performance. We compare the wall time of our methods with exact methods and an approximate MC method. We show a considerable speed-up that scales favorably with the number of objectives. In addition, to evaluate the useability of our approach, we use DeepHV in the setting of MO BO and EAs on a range of synthetic and real-world test functions, and compare with state-of-the-art methods and show competitive performance.

## 2 THE HYPERVOLUME

In this section, we introduce relevant information regarding the definition and symmetry properties of the hypervolume, which will later complement the definition of our method.

**Hypervolume - Definition**   For a natural number $N \in \mathbb{N}$, we put $[N] := \{1, \ldots, N\}$. Consider (column) vectors $\mathbf{y}_n \in \mathbb{R}^M$ in the $M$-dimensional Euclidean space. For a subset $J \subseteq \mathbb{N}$, we abbreviate: $\mathbf{y}_J := [\mathbf{y}_j | j \in J]^\top \in \mathbb{R}^{M \times |J|}$, so $\mathbf{y}_{[N]} := [\mathbf{y}_1, \ldots, \mathbf{y}_N] \in \mathbb{R}^{M \times N}$. We then define the ($M$-dimensional) *hypervolume* of $\mathbf{y}_J$ w.r.t. a fixed reference point $\mathbf{r} \in \mathbb{R}^M$ as:

$$\mathrm{HV}_\mathbf{r}^M(\mathbf{y}_J) := \mathrm{HV}_\mathbf{r}^M(\mathbf{y}_{j_1}, \ldots, \mathbf{y}_{j_{|J|}}) := \boldsymbol{\lambda}^M \left( \bigcup_{j \in J} [\mathbf{r}, \mathbf{y}_j] \right), \tag{5}$$

where $\boldsymbol{\lambda}^M$ is the $M$-dimensional Lebesgue measure, which assigns to a subset of $\mathbb{R}^M$ their $M$-dimensional (hyper) volume, and where we for $\mathbf{y} = [y_1, \ldots, y_M]^\top \in \mathbb{R}^M$ abbreviated the $M$-dimensional cube as: $[\mathbf{r}, \mathbf{y}] := [r_1, y_1] \times \cdots \times [r_M, y_M]$, if $r_m \leq y_m$ for all $m \in [M]$, and the empty-set otherwise. For fixed $N$ and another point $\mathbf{y} \in \mathbb{R}^M$ we define its *hypervolume improvement* (over $\mathbf{y}_{[N]}$) as:

$$\mathrm{HVI}(\mathbf{y}) := \mathrm{HVI}_\mathbf{r}^M(\mathbf{y}|\mathbf{y}_{[N]}) := \mathrm{HV}_\mathbf{r}^M(\mathbf{y}, \mathbf{y}_1, \ldots, \mathbf{y}_N) - \mathrm{HV}_\mathbf{r}^M(\mathbf{y}_1, \ldots, \mathbf{y}_N). \tag{6}$$

With help of the *inclusion-exclusion formula* we can explicitely rewrite the hypervolume as:

$$\mathrm{HV}_\mathbf{r}^M(\mathbf{y}_J) = \sum_{\emptyset \neq S \subseteq J} (-1)^{|S|+1} \cdot \prod_{m=1}^M \min_{s \in S} (y_{m,s} - r_m)_+. \tag{7}$$

**The Symmetry Properties of the Hypervolume Function**   To state the symmetry properties of the hypervolume function, we will fix our reference point $\mathbf{r} := \mathbf{0}$, which is w.l.o.g. always possible by subtracting $\mathbf{r}$ from each $\mathbf{y}_n$.

The hypervolume then scales with each of its components. So for $\mathbf{c} = [c_1, \ldots, c_M]^\top \in \mathbb{R}_{>0}^M$:

$$\mathrm{HV}^M(\mathbf{c} \odot \mathbf{y}_{[N]}) = c_1 \cdots c_M \cdot \mathrm{HV}^M(\mathbf{y}_{[N]}), \tag{8}$$

where $\mathbf{c} \odot \mathbf{y}_{[N]}$ means that the $m$-th entry of each point $\mathbf{y}_n \in \mathbb{R}_{\geq 0}^M$ is multiplied by $c_m$ for all $m \in [M]$. Furthermore, the order of the points does not change the hypervolume:

$$\mathrm{HV}^M(\mathbf{y}_{\sigma(1)}, \ldots, \mathbf{y}_{\sigma(N)}) = \mathrm{HV}^M(\mathbf{y}_1, \ldots, \mathbf{y}_N), \tag{9}$$

for every permutation $\sigma \in \mathbb{S}_N$, where the latter denotes the *symmetric group*, the group of all permutations of $N$ elements. But also the order of the $M$-components does not change the hypervolume:

$$\mathrm{HV}^M(\tau \odot \mathbf{y}_{[N]}) = \mathrm{HV}^M(\mathbf{y}_{[N]}), \tag{10}$$

for every permutation $\tau \in \mathbb{S}_M$, where $\tau \odot \mathbf{y}_{[N]}$ means that the row indices are permuted by $\tau$.

To jointly formalize these equivariance properties of $\mathrm{HV}^M$ together, we define the following joint group as the semi-direct product of scalings and permutations:

$$G := \mathbb{R}_{>0}^M \rtimes \mathbb{S}_M \times \mathbb{S}_N, \tag{11}$$

where the group operation is defined as follows on elements $(\mathbf{c}_1, \tau_1, \sigma_1)$ and $(\mathbf{c}_2, \tau_2, \sigma_2) \in G$ via:

$$(\mathbf{c}_2, \tau_2, \sigma_2) \cdot (\mathbf{c}_1, \tau_1, \sigma_1) := (\mathbf{c}_2 \cdot \mathbf{c}_1^{\tau_2}, \tau_2 \circ \tau_1, \sigma_2 \circ \sigma_1) \in G, \tag{12}$$

where with $\mathbf{c}_1 = [c_{1,1}, \ldots, c_{M,1}]^\top \in \mathbb{R}_{>0}^M$ we use the abbreviation:

$$\mathbf{c}_1^{\tau_2} := [c_{\tau_2(1),1}, \ldots, c_{\tau_2(M),1}]^\top \in \mathbb{R}_{>0}^M. \tag{13}$$

This group then acts on the space of non-negative $(M \times N)$-matrices as follows. For $\mathbf{Y} = (y_{m,n})_{m,n} \in \mathbb{R}_{\geq 0}^{M \times N}$ and $(\mathbf{c}, \tau, \sigma) \in G$ we put:

$$(\mathbf{c}, \tau, \sigma) \odot \mathbf{Y} := \begin{bmatrix} c_1 \cdot y_{\tau(1),\sigma(1)} & \cdots & c_1 \cdot y_{\tau(1),\sigma(N)} \\ \vdots & \ddots & \vdots \\ c_M \cdot y_{\tau(M),\sigma(1)} & \cdots & c_M \cdot y_{\tau(M),\sigma(N)} \end{bmatrix}. \tag{14}$$

The mentioned symmetries of the hypervolume can then be jointly summarized by the following *G-equivariance* property:

$$\mathrm{HV}^M((\mathbf{c}, \tau, \sigma) \odot \mathbf{Y}) = c_1 \cdots c_M \cdot \mathrm{HV}^M(\mathbf{Y}). \tag{15}$$

This is the property that we will exploit for an efficient approximation of the hypervolume with a $G$-equivariant deep neural network.

The hypervolume for different dimensions $M$ are related as follows:

$$\mathrm{HV}^M(\mathbf{y}_J) = \mathrm{HV}^{M+K}\left( \begin{bmatrix} \mathbf{y}_J \\ \mathbf{1}_{K \times |J|} \end{bmatrix} \right), \tag{16}$$

where we padded the $(K \times |J|)$-matrix with ones to $\mathbf{y}_J$, or, equivalently, $K$ ones to each point $\mathbf{y}_j$, $j \in J$. We leverage this property to combine datasets with different dimensions $M$ and to train a model that can generalize to multiple dimensions.

## 3 DeepHV - Equivariant Deep Hypervolume Approximation

In this section, we first present the $G$-equivariant neural network layers used in DeepHV that incorporates the symmetry properties defined in Sec. 2. Following this, we discuss the general architecture of DeepHV. We provide an intuitive explanation and mathematical proofs in App. A.1.

### 3.1 The Equivariant Layers

Before we introduce our $G$-equivariant neural network layers, we first recall how $(\mathbb{S}_M \times \mathbb{S}_N)$-equivariant layers are constructed.

**Theorem 1** (See Hartford et al. (2018) Thm. 2.1). *A fully connected layer of a neural network* $\mathrm{vec}(\mathbf{Z}) = \sigma(\mathbf{W} \, \mathrm{vec}(\mathbf{Y}))$ *with input matrix* $\mathbf{Y} \in \mathbb{R}^{M \times N}$, *output matrix* $\mathbf{Z} \in \mathbb{R}^{M \times N}$, *weight matrix* $\mathbf{W} \in \mathbb{R}^{(MN) \times (MN)}$ *and strictly monotonic activation function* $\sigma$ *is* $(\mathbb{S}_M \times \mathbb{S}_N)$-equivariant if and only if it can be written as:*

$$\mathbf{Z} = \sigma\left( w_1 \cdot \mathbf{Y} + w_2 \cdot \mathbf{1}_M \cdot \mathbf{Y}_{\mathbb{M}:} + w_3 \cdot \mathbf{Y}_{:\mathbb{M}} \cdot \mathbf{1}_N^\top + w_4 \cdot \mathbf{Y}_{\mathbb{M}:\mathbb{M}} \cdot \mathbf{1}_{M \times N} + w_5 \cdot \mathbf{1}_{M \times N} \right), \tag{17}$$

*where* $\mathbf{1}$ *denotes the matrix of ones of the indicated shape,* $w_1, \ldots, w_5 \in \mathbb{R}$ *and where* $\mathbb{M}$ *is the mean operation, applied to rows* $(\mathbf{Y}_{\mathbb{M}:})$, *columns* $(\mathbf{Y}_{:\mathbb{M}})$ *or both* $(\mathbf{Y}_{\mathbb{M}:\mathbb{M}})$.

We define the *row scale* of a matrix $\mathbf{Y} = (y_{m,n})_{m,n} \in \mathbb{R}^{M \times N}$ via:

$$\mathbf{s}(\mathbf{Y}) := \left[ \max_{n \in [N]} |y_{m,n}| \,\middle|\, m \in [M] \right]^{\top} \in \mathbb{R}^M, \tag{18}$$

and its row-wise inverse $\mathbf{s}^{-1}(\mathbf{Y}) \in \mathbb{R}^M$. We then abbreviate the *row-rescaled* matrix as:

$$\mathbf{Y}_{\oslash} := \mathbf{s}^{-1}(\mathbf{Y}) \odot \mathbf{Y} \in \mathbb{R}^{M \times N}. \tag{19}$$

With these notations, we are now able to construct a $G$-equivariant neural network layer with input channel indices $i \in I$ and output channel indices $o \in O$:

$$\mathbf{Z}^{(o)} = \sigma_\alpha \Bigg( \frac{1}{|I|} \sum_{i \in I} \mathbf{s} \left( \mathbf{Y}^{(i)} \right) \odot \left( w_1^{(o,i)} \cdot \left( \mathbf{Y}_{\oslash}^{(i)} \right) + w_2^{(o,i)} \cdot \mathbf{1}_M \cdot \left( \mathbf{Y}_{\oslash}^{(i)} \right)_{\mathbb{M}:} \right. \tag{20}$$

$$\left. + w_3^{(o,i)} \cdot \left( \mathbf{Y}_{\oslash}^{(i)} \right)_{:\mathbb{M}} \cdot \mathbf{1}_N^{\top} + w_4^{(o,i)} \cdot \left( \mathbf{Y}_{\oslash}^{(i)} \right)_{\mathbb{M}:\mathbb{M}} \cdot \mathbf{1}_{M \times N} + w_5^{(o)} \cdot \mathbf{1}_{M \times N} \right) \Bigg),$$

where $\mathbb{M}$ is applied to the row-rescaled $\mathbf{Y}$ row-wise ($\mathbf{Y}_{\mathbb{M}:}$), column-wise ($\mathbf{Y}_{:\mathbb{M}}$) or both ($\mathbf{Y}_{\mathbb{M}:\mathbb{M}}$), and where, again, $\mathbb{M}$ denotes the mean operation (but could also be max or min), and weights $w_k^{(o,i)}, w_5^{(o)} \in \mathbb{R}$ for $k \in [4]$, $o \in O$, $i \in I$. $\sigma_\alpha$ denotes any homogeneous activation function, i.e. we require that for every $y \in \mathbb{R}$, $c \in \mathbb{R}_{>0}$:

$$\sigma_\alpha(c \cdot y) = c \cdot \sigma_\alpha(y) \tag{21}$$

This is, for instance, met by the leaky-ReLU activation function. An alternative way to achieve equivariance of the whole network w.r.t. to the scale of the whole network is by extracting the scales at the input layer and then multiplying them all against the output layer, in contrast to propagating the scale through every layer individually.

## 3.2 ARCHITECTURE

All the DeepHV models presented here comprised of the same architecture, with the exception that we change the number of input and output channels in order to trade off the number of model parameters and the expressivity of the model. We denote models as DeepHV-$c$, $c$ denoting the number of channels. Before the $\mathbb{R}^{N \times M}$ input is passed through the layers, we first extract and store the scale using eq. 18, and scale the inputs using eq. 19. We use five equivariant layers as in eq. 20, where the first layer has 1 input channel and $c$ output channels. The three intermediate layers have the same number $c$ of input and output channels. The last layer then maps back to 1 output channel, again resulting into $\mathbb{R}^{N \times M}$ output. Consequently, the mean is taken of the resulting output to bring it back to a permutation invariant quantity ($\mathbb{R}^1$). This is then passed through a sigmoid function to map values between 0 and 1, as this is the value which the hypervolume can have in the scaled setting. Finally, we rescale the output by multiplying it with the product of the scaling factors. This step ensures that the model output is scale equivariant.

## 4 TRAINING

### 4.1 DATA GENERATION

To generate training data, we adopt a similar strategy as proposed in Shang et al. (2022a), which is defined as follows: 1. Uniformly sample an integer $n \in [1, 100]$; 2. Uniformly sample 1000 solutions in $[0, 1]^M$; 3. Obtain different fronts $\{P_1, P_2, ...\}$ using non-dominated sorting, where $P_1$ is the first Pareto front and $P_2$ is the following front after all solutions of $P_1$ are removed; 4. Identify all fronts $P_i$ with $|P_i| \geq n$. If no front satisfies this conditions, go back to Step 2; 5. Randomly select one front $P_i$ with $|P_i| \geq n$ and randomly select $n$ solutions from the front to construct one solution set. Using this procedure, we generate datasets consisting of 1 million solution sets for each of the objective cases $3 \leq M \leq 10$. We use this procedure as it generates a wide variety of non-dominated solution sets (see Shang et al. (2022a)) and so that we can compare our model performance with HV-Net. We split our datasets into 800K training points and 100K validation and test points, respectively. We note that as we only sample solution sets with at maximum a hundred solutions, our model is likely restricted to this setting. In addition, using the relation of eq. 16, we also pad all datasets of $M < 10$ to $M = 10$, by padding it with ones. As this does not change the hypervolume values of each solution set, we create a universal dataset that incorporates all generated objective cases.

## 4.2 PERFORMANCE

We train models with 64, 90, 128, and 256 channels (See Sec. 3.2 for architecture) on each objective case separately. In addition, we also train models on the combined dataset (See Sec. 4.1, denoted with -all) using 128 and 256 channels. All DeepHV models have been trained with a learning rate of $10^{-5}$, using Adam and the Mean Absolute Percentage Er-

Table 1: Test MAPE (lower is better) for models trained on each objective $M$ separately. Bold numbers indicate the best-performing model. Blue numbers indicate the best model between HV-Net (Shang et al., 2022a) and our models with a comparable number of model parameters.

| M | DeepHV-64 | DeepHV-90 | DeepHV-128 | DeepHV-256 | HV-Net |
|---|-----------|-----------|------------|------------|--------|
| 3 | 0.00800 | 0.00744 | 0.00526 | **0.00483** | 0.015878 |
| 5 | 0.02065 | 0.01608 | 0.01689 | **0.01209** | 0.032067 |
| 8 | 0.04710 | 0.03381 | 0.01506 | **0.01093** | 0.042566 |
| 10 | 0.03203 | 0.02783 | 0.02395 | **0.01148** | 0.050867 |

ror (MAPE) loss function (de Myttenaere et al., 2016). As the (scaled) hypervolume tends to become small in high-dimension cases, more common loss functions such as the RMSE become rather small rather quickly, despite having large relative errors. The MAPE is better suited for this setting. For the separate models, we use a batch size of 64 and train for 200 epochs. However, improvements where typically marginal after $\sim 60$ epochs. For the models trained on all objective cases simultaneously, we train for 100 epochs with a batch size of 128. We pick the best model within these epochs based on the lowest loss on the validation partition. We report MAPE test errors in Table 1 and compare with HV-Net (Shang et al., 2022a). Note that HV-Net uses 1 million training datapoints, whereas we use 800K. When using 90 channels (DeepHV-90), our model has a comparable number of parameters (98.3K) w.r.t HV-Net (99.7K), however we obtain roughly a twofold increase of performance on most objective cases. In fact, using 64 channels (49K parameters), we already outperform HV-Net on most objective cases. This boost in performance is likely due to the incorporation of scale equivariance and an additional permutation invariance over the order of the objectives, which are not incorporated into HV-Net. For larger models, we obtain even better results.

We report results on DeepHV trained on all objective cases simultaneously in Table 2. Here it is shown that the general models have a significant increase in performance compared to the separate models. Interestingly, this shows that the model can generalize to multiple objective and can gain additional performance in each objective case separately by seeing instances of others. This potentially allows for the training of a universal model that can be trained on even higher objective cases.

Table 2: Test MAPE for models trained on each objective case separately, and for models trained on all objective cases simultaneously (denoted with additional -all). Bold numbers indicate the best model in their architecture.

| M | DeepHV-128 | DeepHV-128-all | DeepHV-256 | DeepHV-256-all |
|---|-----------|----------------|------------|----------------|
| 3 | **0.00526** | 0.00673 | **0.00483** | 0.00495 |
| 4 | 0.00964 | **0.00887** | 0.00782 | **0.00714** |
| 5 | 0.01689 | **0.00956** | 0.01209 | **0.00773** |
| 6 | 0.01511 | **0.00954** | 0.01354 | **0.00761** |
| 7 | 0.01164 | **0.00900** | 0.00967 | **0.00707** |
| 8 | 0.01506 | **0.00849** | 0.01093 | **0.00657** |
| 9 | 0.01506 | **0.00796** | 0.01216 | **0.00601** |
| 10 | 0.02395 | **0.00772** | 0.01148 | **0.00598** |

## 4.3 ABLATION STUDY

To further study the effect of the symmetries incorporated into DeepHV, we perform an ablation study. We train different models to address the cases of: a) no permutation invariances and no scaling equivariance; b) no permutation invariances, but with scaling equivariance; c) only permutation invariances but no scaling equivariance; and d) both permutation invariances and scaling equivariance. For the case a) we use a standard multi-layer perceptron (MLP). For case b) we use an MLP with scaled inputs (using eq. 18 and eq. 19), followed by rescaling of the model outputs. For case c), we use the DeepHV architecture without scale equivariance. Case d) corresponds to the original DeepHV architecture. All models had a similar number of parameters and were kept as comparable to each other as possible, details can be found in App. B. Without incorporating scale equivariance, training with the MAPE loss was unstable and got stuck around a value of 1, meaning that the model always outputs hypervolumes close to zero, this was also observed in Shang et al. (2022a). We therefore also

performed training with the MSE loss and log MSE loss and report the best model found using these loss functions. We report MAPE test errors in Tab. 3, where it is shown that each additional symmetry leads to increased model performance. This becomes especially profound at higher objective cases $M > 5$, where the MLPs do not achieve lower errors than 19%.

Table 3: Test Mean Absolute Percentage Error. Bold numbers indicate the best-performing model.

| M | MLP | MLP-with-scaling | DeepHV-128-no-scaling | DeepHV-128 |
|---|---|---|---|---|
| 3 | 0.03103 | 0.02397 | 0.01944 | **0.00526** |
| 5 | 0.06483 | 0.05820 | 0.03311 | **0.01689** |
| 8 | 0.2015 | 0.1928 | 0.04571 | **0.01506** |
| 10 | 0.3106 | 0.3031 | 0.05212 | **0.02395** |

## 5 EXPERIMENTS

### 5.1 TIME COMPARISON

We compare the time performance of DeepHV with the exact hypervolume methods of Lacour et al. (2017) (complexity $O(N^{M/2+1})$) implemented in BoTorch (Balandat et al., 2020) and the method of Fonseca et al. (2006) (complexity ($O(N^{M-2} \log N)$) as implemented in Pymoo (Blank & Deb, 2020). In addition, we compare time performance with an approximate Monte Carlo (MC) hypervolume method implemented in Pymoo. We perform the tests on CPU and GPU where applicable. Experiments shown in Fig. 2 comprised of the computation of 100 solution sets randomly drawn from the test sets. The MC method used 10K samples. In all cases, DeepHV is faster than the BoTorch and Pymoo MC implementation. Above $M = 5$, both variants of DeepHV are also faster than the exact Pymoo implementation. We also compare the MAPE versus the computation time for DeepHV and the MC method with different number of samples in App. D.1.

### 5.2 BENCHMARKS

We empirically evaluate DeepHV on a range synthetic test problems from the DTLZ (Deb et al., 2002b) problem suite, which contains standard test problems from the MO optimization literature that are scalable in the number of input variables $d$ and objectives $M$ and contain vastly different types of Pareto fronts. In Sec. 5.2.1, we test DeepHV in the context of multi-objective evolutionary algorithms (MO EAs) and in Sec. 5.2.2 in the context of Bayesian optimization (BO). In addition, we consider a range of challenging real-world problems (Tanabe & Ishibuchi, 2020): a penicillin production ($M = 3$) problem, a vehicle safety problem ($M = 3$), a car side impact problem ($M = 4$), a water resource planning problem ($M = 6$), and a conceptual marine design problem ($M = 4$). See App. C.2 for more details regarding the problems and reference point specifications. We run all methods five times and report the means and two standard errors. Problems not reported in the consecutive sections are shown in in Sec. D.2.2 and D.3.2.

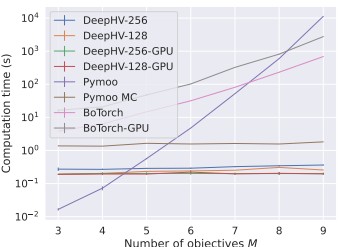

Figure 2: Comparison of computation time, shown on log-scale. We report the means and standard errors across 3 trials.

### 5.2.1 MULTI-OBJECTIVE EVOLUTIONARY ALGORITHMS

To use DeepHV in the context of MO EAs, we replace the exact computation of the hypervolume in SMS-EMOA (Beume et al., 2007) with that of DeepHV, and compare performance. In addition, we compare performance with NSGA-II (Deb et al., 2002a) and NSGA-III (Deb & Jain, 2014). SMS-EMOA (Beume et al., 2007) is an EA that uses the hypervolume to guide optimization. In its original form, it uses a steady-state ($\mu + 1$) approach to evolve its population, where at each generation, one offspring is created and added to the population. Next, one solution is removed from the population by first ranking the population into separate fronts, followed by the removal of the

solution that has the least contribution to the hypervolume of its respective front. This is done by computing the hypervolume contribution of each solution, by computing the hypervolume difference between the respective front, and the front without the individual solution. Instead of a steady-state approach, we use a $(\mu + \mu)$ approach to evolve the population, effectively doubling the population at each generation, and then need to select half of the population using the selection strategy described above. We stress that this is an extensive test of the performance of DeepHV, as the model needs to predict both the correct hypervolume of the front, but also the contribution of each solution in the front (i.e. the front and all of its sub-fronts need to be predicted with high accuracy). In the case of $M > 5$, we use the MC approximation of the hypervolume instead of the exact hypervolume as its use becomes computationally prohibitive (See Fig. 2). We note that the DeepHV architecture could also be adopted and trained to directly predict the hypervolume contribution of each solution in the solution as in (Shang et al., 2022b), which would be more suitable for this task.

As another baseline, we compare with NSGA-II which is a renowned MO EA, due to its low computational overhead. Although it generally is less sample-efficient than SMS-EMOA, it is often the method of choice due to a lack of scalable alternatives. In NSGA-II, the population is evolved using a $(\mu + \mu)$ approach, then individuals are first selected front wise using non-dominated sorting and then, remaining individuals are selected based on the Manhattan distance in the objective space, in order to maintain a diverse population. The task of the Manhattan distance corresponds to that of the hypervolume contribution in SMS-EMOA. NSGA-III replaces the Manhattan distance with a set of reference directions, where population is selected at each reference direction to maintain diversity, but is otherwise similar to NSGA-II. For each algorithm, we use a population size of 100 individuals and run for 100 generations, resulting into 10K function evaluations, and we record the exact hypervolume of each generation. Further algorithmic details can be found in the App. C.1.

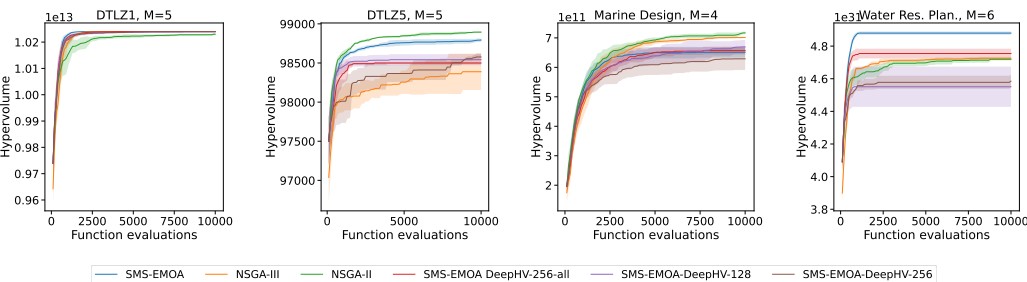

Figure 3: Optimization performance in terms of exact hypervolume (higher is better) versus function evaluations.

Fig. 3 presents results on the DTLZ1 and DTLZ5 problems for $M = 5$, and on the real-world marine design and water resource planning. We report wall times in Sec. D.2.1, and report results on additional synthetic test problems in App. D.2.2. The Pareto front of DTLZ1 lies on a the linear hyperplane $\sum_{m=1}^{M} f_m^* = 0.5$, and all methods find similar hypervolume, with a slightly worse performance of NSGA-II. DTLZ5's has a beam-like Pareto front. On this problem, NSGA-II and SMS-EMOA perform best, followed by the DeepHV models. NSGA-III performs worst. On the Marine Design problem, the NSGA variations perform best, followed by SMS-EMOA and the DeepHV variations which perform similar. On the water resource planning problem, SMS-EMOA performs best, followed by DeepHV-256-all and the NSGA variations. Although the different DeepHV models perform similar on most problems, on this problem the separately trained models perform worst.

The closeness of the DeepHV variations to the exact SMS-EMOA provides an indication on the ability of the models to generalize to the different encountered Pareto front shapes. We observe that on most problems, exact SMS-EMOA (blue line) typically obtains higher hypervolumes or similar than the DeepHV variations. On problems where SMS-EMOA performs better, it is observed that that DeepHV finds important points on the pareto front, but is not able to spread the population over the Pareto front as well as SMS-EMOA.

### 5.2.2 MULTI-OBJECTIVE BAYESIAN OPTIMIZATION

We use DeepHV in the context of MO BO by using DeepHV to compute the hypervolume improvement (HVI, eq. 6). We use the HVI in a Upper Confidence Bound (UCB) acquisition function, which

we either name UCB-HV when the exact hypervolume is used, or UCB-DeepHV in case we use DeepHV to approximate the hypervolume. It is defined as follows:

$$\alpha_{\text{UCB}-\text{DeepHV}}(\mathbf{x}, \beta) = \text{HVI}(\mathbf{f}_{\text{UCB}}(\mathbf{x}, \beta)), \text{ where } \mathbf{f}_{\text{UCB}}(\mathbf{x}, \beta) = \mu(\mathbf{x}) + \beta\sigma(\mathbf{x}). \quad (22)$$

Here, $\mu(\mathbf{x})$ and $\sigma(\mathbf{x})$ are the mean and standard deviation of the GP posterior at input $\mathbf{x}$, whereas $\beta$ is a trade-off parameter that controls exploration and exploration. In this work we use $\beta = 0.3$. We compare with the state-of-the-art methods qParEgo and qEHVI (Daulton et al., 2020). ParEgo (Knowles, 2006) randomly scalarizes the objectives and uses the Expected Improvement acquisition function (Jones et al., 1998). qParego uses an MC-based Expected Improvement acquisition function and uses exact gradients via auto-differentiation for acquisition optimization. Expected Hypervolume Improvement (EHVI) is the expectation of the HVI (Yang et al., 2019), thus integrating the HVI over the Gaussian process posterior. qEHVI is a parallel extension of EHVI and allows for acquistion optimization using auto-differentation and exact gradients, making it more computationally tractable than EHVI. qEHVI was empirically shown to outperform many other MO BO methods (including qParEgo) (Daulton et al., 2020; 2022). We evaluate optimization performance in terms of the exact hypervolume on four test problems. Note that above the five objectives case, qEHVI becomes computationally prohibitive. UCB-HV becomes computationally prohibitive for $M > 4$. We show wall-clock runtime information in Sec. D.3.1 and we provide additional experiments on other objective cases and additional problems in Sec. D.3. Fig. 4 shows results for a range of DTLZ synthetic test problems and the Car Side Impact problem. In general, it is expected that qEHVI is superior to UCB-HV and UCB-DeepHV, as it uses the exact hypervolume and integrates out the uncertainty of the GP posterior in a more elaborate way. However, if these methods perform on par with qEHVI, then the acquisition function choice is reasonable. If UCB-DeepHV performs similarly to UCB-HV, it would that DeepHV approximated the hypervolume of the test problems well. qEHVI, ranks best on DTLZ2, DTLZ7 and the Car Side Impact problem. On DTLZ5, qEHVI performs worse than than the other methods. On most problems, UCB-HV and UCB-DeepHV have similar performance, and generally outperform qParEgo, which is a strong result.

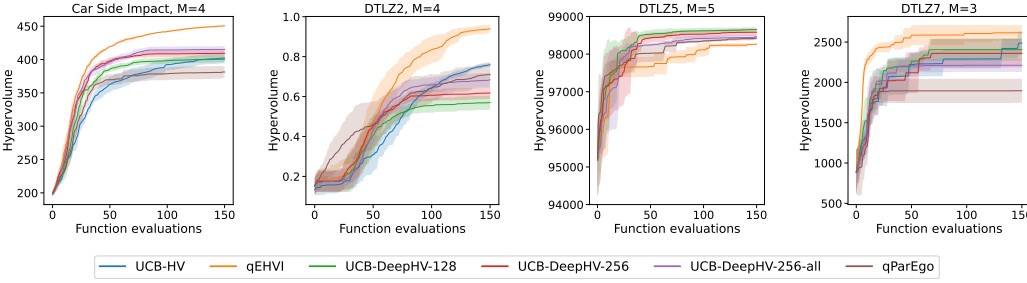

Figure 4: Sequential optimization performance in terms of hypervolume versus function evaluations.

## 6 DISCUSSION

We presented DeepHV, a novel hypervolume approximation based on deep learning. DeepHV is scale equivariant in each of the objectives as well as permutation invariant w.r.t. both the objectives and the samples, thus incorporating important symmetry properties of the hypervolume. We showed that our method obtains lower approximation errors compared to HV-Net that does not incorporate all of these symmetries. In addition, we showed through an ablation study that incorporating these symmetries leads to improved model performance. We showed that DeepHV achieved lower approximation errors than a MC based approximation and is considerably faster than exact methods, becoming more pronounced at high objective cases $M > 4$. Therefore, we believe DeepHV is promising as a replacement for computationally demanding exact hypervolume computations in state-of-the-art (high-throughput) methods. A limitation is that the DeepHV models are currently only trained on solution sets with at maximum 100 non-dominated solutions, restricting their use to this setting. However it can handle arbitrary amount of objectives and samples by design. We tested DeepHV in context of multi-objective BO and EAs and we showed competitive performance compared with state-of-the-art methods. We believe these results are promising, and with better training data generation methods, DeepHV could potentially be even more competitive with exact hypervolume methods.

## 7 ETHICS STATEMENT

The paper presents a method that can be used in multi-objective optimization problems. These problems find widespread use throughout society. For example in material design (catalysts, medicines, etc.), car design or finance. Although many such applications are for the general good, there are also applications which can have ethically debatable motivations, for instance, when used in the design of toxins, narcotics, or weaponry.

## 8 REPRODUCIBILITY STATEMENT

Code, models, and datasets used in this work can be found at: `https://github.com/Jimbo994/deephv-iclr`. Where applicable, we have run experiments multiple times with different random seeds and have stated mean and standard error of the results. In addition, we have described important implementation and experimental details in either the main body of the text or the Appendix.

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

# A PROOFS

**Intuitive explanation** We here provide an intuitive explanation which complements the mathematics in Sec 3.1 and the provided proof in the next paragraph. Consider a matrix $\mathbf{Y} \in \mathbb{R}^{M \times N}$ matrix to which we want to apply operations which are both permutation equivariant or invariant w.r.t. $M$ and $N$ and again result into an $\mathbb{R}^{M \times N}$ output. The simplest operation that is invariant would be to apply a mean operation to both the columns ($M$) and rows ($N$) and multiplying by $\mathbf{1}_{M \times N}$ (to maintain an $\mathbb{R}^{M \times N}$ output). However, other invariant operations such as taking the min, max or sum would also be possible. Note that the resulting invariant output can also be multiplied with scalar values (i.e.,

model weights) without violating the permutation invariance. This is the fourth term in eq. 20 (with $w_4$). The simplest equivariant operation would be to multiply $\mathbf{Y}$ with a scalar value, which is the first term in eq. 20 (with $w_1$). Shifting all $\mathbb{R}^{M \times N}$ values with some bias term, does not break this equivariance, this is the fifth term in eq. 20 (with $w_5$). In similar fashion, we can take the mean over the $M$ rows and multiply with $\mathbf{1}_{M \times N}$, which would result in an output that is equivariant w.r.t. to the columns and invariant w.r.t. the rows, this can then also be multiplied with a scalar value ($w_2$ in in eq. 20). Likewise, we can do the same thing over the columns, which is the third term in eq. 20 (with $w_3$). In addition, we would like to do transformations which are scaling equivariant. This can be done by rescaling the inputs to some domain, and then rescaling the outputs after these operations.

### A.1 Proof of Section 3.1

**Proof** To show that the layers in eq. 20 are really $G$-equivariant, let $\mathbf{Y}^I := (\mathbf{Y}^{(i)})_{i \in I}$ and $g(\mathbf{Y}^I)$ the left hand side of eq. 20, and $(\mathbf{c}, \tau, \sigma) \in G$. Then $(\mathbf{c}, \tau, \sigma) \odot \mathbf{Y}^I = ((\mathbf{c}, \tau, \sigma) \odot \mathbf{Y}^{(i)})_{i \in I}$. We can also see that:

$$
\mathbf{s}((\mathbf{c}, \tau, \sigma) \odot \mathbf{Y}^{(i)}) = \left[ \max_{n \in [N]} |c_m \cdot y_{\tau(m), \sigma(n)}| \,\middle|\, m \in [M] \right]^\top \tag{23}
$$

$$
= \left[ c_m \cdot \max_{n \in [N]} |y_{\tau(m), n}| \,\middle|\, m \in [M] \right]^\top = \mathbf{c} \cdot \mathbf{s}(\mathbf{Y}^{(i)})^\tau, \tag{24}
$$

where $\sigma$ acts trivially on the $(M \times 1)$-vector $\mathbf{s}(\mathbf{Y}^{(i)})$. If we act on the defining equation for $\mathbf{Y}_{\oslash}^{(i)}$:

$$
\mathbf{Y}^{(i)} = \mathbf{s}(\mathbf{Y}^{(i)}) \odot \mathbf{Y}_{\oslash}^{(i)}, \tag{25}
$$

with $(\mathbf{c}, \tau, \sigma)$ we then get:

$$
(\mathbf{c}, \tau, \sigma) \odot \mathbf{Y}^{(i)} = (\mathbf{c}, \tau, \sigma) \odot \left( \mathbf{s}(\mathbf{Y}^{(i)}) \odot \mathbf{Y}_{\oslash}^{(i)} \right) \tag{26}
$$

$$
= (\mathbf{c} \cdot \mathbf{s}(\mathbf{Y}^{(i)})^\tau, \tau, \sigma) \odot \mathbf{Y}_{\oslash}^{(i)} \tag{27}
$$

$$
= (\mathbf{s}((\mathbf{c}, \tau, \sigma) \odot \mathbf{Y}^{(i)}), \tau, \sigma) \odot \mathbf{Y}_{\oslash}^{(i)} \tag{28}
$$

$$
= \mathbf{s}((\mathbf{c}, \tau, \sigma) \odot \mathbf{Y}^{(i)}) \odot (\tau, \sigma) \odot \mathbf{Y}_{\oslash}^{(i)}. \tag{29}
$$

This shows that:

$$
\left( (\mathbf{c}, \tau, \sigma) \odot \mathbf{Y}^{(i)} \right)_{\oslash} = (\tau, \sigma) \odot \mathbf{Y}_{\oslash}^{(i)}. \tag{30}
$$

With this we get:

$$
h((\mathbf{c}, \tau, \sigma) \odot \mathbf{Y}^{(i)}) \tag{31}
$$

$$
:= w_1^{(o,i)} \cdot \left( (\mathbf{c}, \tau, \sigma) \odot \mathbf{Y}^{(i)} \right)_{\oslash} + w_2^{(o,i)} \cdot \mathbf{1}_M \cdot \left( \left( (\mathbf{c}, \tau, \sigma) \odot \mathbf{Y}^{(i)} \right)_{\oslash} \right)_{\mathbb{M}:} \tag{32}
$$

$$
+ w_3^{(o,i)} \cdot \left( \left( (\mathbf{c}, \tau, \sigma) \odot \mathbf{Y}^{(i)} \right)_{\oslash} \right)_{:\mathbb{M}} \cdot \mathbf{1}_N^\top \tag{33}
$$

$$
+ w_4^{(o,i)} \cdot \left( \left( (\mathbf{c}, \tau, \sigma) \odot \mathbf{Y}^{(i)} \right)_{\oslash} \right)_{\mathbb{M}:\mathbb{M}} \cdot \mathbf{1}_{M \times N} + w_5^{(o,i)} \cdot \mathbf{1}_{M \times N} \tag{34}
$$

$$
= w_1^{(o,i)} \cdot (\tau, \sigma) \odot \mathbf{Y}_{\oslash}^{(i)} + w_2^{(o,i)} \cdot (\tau, \sigma) \odot \left( \mathbf{1}_M \cdot \left( \mathbf{Y}_{\oslash}^{(i)} \right)_{\mathbb{M}:} \right) \tag{35}
$$

$$
+ w_3^{(o,i)} \cdot (\tau, \sigma) \odot \left( \left( \mathbf{Y}_{\oslash}^{(i)} \right)_{:\mathbb{M}} \cdot \mathbf{1}_N^\top \right) \tag{36}
$$

$$
+ w_4^{(o,i)} \cdot (\tau, \sigma) \odot \left( \left( \mathbf{Y}_{\oslash}^{(i)} \right)_{\mathbb{M}:\mathbb{M}} \cdot \mathbf{1}_{M \times N} \right) + w_5^{(o,i)} \cdot (\tau, \sigma) \odot \mathbf{1}_{M \times N} \tag{37}
$$

$$
= (\tau, \sigma) \odot h(\mathbf{Y}^{(i)}). \tag{38}
$$

With these equations we can finally get:

$$g((\mathbf{c}, \tau, \sigma) \odot \mathbf{Y}^I) = \sigma_\alpha \left( \frac{1}{|I|} \sum_{i \in I} \mathbf{s}\left( (\mathbf{c}, \tau, \sigma) \odot \mathbf{Y}^{(i)} \right) \odot h\left( (\mathbf{c}, \tau, \sigma) \odot \mathbf{Y}^{(i)} \right) \right) \tag{39}$$

$$= \sigma_\alpha \left( \frac{1}{|I|} \sum_{i \in I} \mathbf{c} \cdot \mathbf{s}(\mathbf{Y}^{(i)})^\tau \odot (\tau, \sigma) \odot h\left( \mathbf{Y}^{(i)} \right) \right) \tag{40}$$

$$= \sigma_\alpha \left( \frac{1}{|I|} \sum_{i \in I} (\mathbf{c} \cdot \mathbf{s}(\mathbf{Y}^{(i)})^\tau, \tau, \sigma) \odot h\left( \mathbf{Y}^{(i)} \right) \right) \tag{41}$$

$$= \sigma_\alpha \left( \frac{1}{|I|} \sum_{i \in I} (\mathbf{c}, \tau, \sigma) \odot \mathbf{s}(\mathbf{Y}^{(i)}) \odot h\left( \mathbf{Y}^{(i)} \right) \right) \tag{42}$$

$$= \sigma_\alpha \left( (\mathbf{c}, \tau, \sigma) \odot \frac{1}{|I|} \sum_{i \in I} \mathbf{s}(\mathbf{Y}^{(i)}) \odot h\left( \mathbf{Y}^{(i)} \right) \right) \tag{43}$$

$$= (\mathbf{c}, \tau, \sigma) \odot \sigma_\alpha \left( \frac{1}{|I|} \sum_{i \in I} \mathbf{s}(\mathbf{Y}^{(i)}) \odot h\left( \mathbf{Y}^{(i)} \right) \right) \tag{44}$$

$$= (\mathbf{c}, \tau, \sigma) \odot g(\mathbf{Y}^I). \tag{45}$$

where we in the second to last step made use of the homogeneity of the activation function $\sigma_\alpha$ and that it is applied element-wise. This shows that $g$ is $G$-equivariant.

## B    DETAILS ON ABLATION STUDY

All models were trained with a learning rate of $10^{-5}$ using Adam for 100 epochs and in all layers we use the leaky-ReLu activation function. DeepHV-128 (described in Sec. 3.2) and DeepHV-128-no-scaling were comprised of 198K parameters. The MLPs consisted of 6 layers. The first layer consisted of $M \times 100$ input features and 180 output features. The 4 intermediate layers consisted of 180 input and output features and the last layer consisted of 180 input features and 1 output feature. Similar as for DeepHV this output was then passed through a sigmoid function to map values between 0 and 1. This is also justified for the models without scaling equivariance, as all examples in the train, test, and validation set are between 0 and 1. As $M$ influences the parameters of the first layer, the number of parameters in the model varied per objective case. The MLPs consisted of 184K parameters for $M = 3$, 220K parameters for $M = 5$, 274K parameters for $M = 8$ and 310K parameters for $M = 10$. Even though the MLPs were equipped with more parameters than DeepHV in cases $M > 5$, this did not lead to improved performance.

## C    DETAILS ON EXPERIMENTS

### C.1    ALGORITHMIC DETAILS

**Multi-objective Evolutionary Algorithms:**    For, NSGA-II, NSGA-III, and SMS-EMOA we use the default settings in the open-source implementation of Pymoo [1], where only for SMS-EMOA we replace the hypervolume computation with DeepHV or the MC hypervolume approximation and all other settings are kept default. For the MC hypervolume approximation we use $10,000$ samples.

For NSGA-III, we create reference directions using the das-dennis approach (Das & Dennis, 1998). Where we used the following number of partitions: {M=3:12, M=4:4, M=5:4, M=6:3, M=7:2, M=8:2, M=9:2, M=10:2}

**Multi-objective Bayesian Optimization:**    For all methods and experiments we use an independent Gaussian Process (GP) with a constant mean function and a Matérn-5/2 kernel with automatic relevance detection (ARD) and fit the GP hyperparameters by maximizing the marginal log-likelihood.

---

[1](https://github.com/anyoptimization/pymoo)

All experiments are initialized with $2d + 1$ randomly drawn datapoints. We then allow for a optimization budget of 150 function evaluations. For qEHVI and qParEgo we use 128 quasi-MC samples. All methods are optimized using L-BFGS-B with 10 restarts and 64 raw samples (according to BoTorch naming convention).

**Methods using DeepHV:** We ensure that solution sets passed to DeepHV only contain non-dominated solutions and are shifted so that the reference point lies at $[0]^M$. This is done by subtracting the specified reference point from the solution set and applying non-dominated sorting to only keep the non-dominated solutions.

### C.2 SYNTHETIC & REAL-WORLD PROBLEMS

Information on the reference points used for hypervolume computation of each problem can be found in Table 4 and are based on suggested values in Pymoo (Blank & Deb, 2020) and BoTorch (Balandat et al., 2020) or specified later. The DTLZ problems are minimization problems. As BoTorch assumes maximization, we multiply the objectives and reference points for all synthetic problems by -1 and maximize the resulting objectives.

In the multi-objective Bayesian optimization problems we use the implementations of the synthetic functions as provided in Botorch (Balandat et al., 2020). Whereas in the multi-objective evolutionary algorithm experiments we use the implementations of Pymoo (Blank & Deb, 2020).

**DTLZ:** The DTLZ test suite contains a range of test functions which are considered standard test problems in the multi-objective optimization literature. Mathematical formulas of each test function are provided in Deb et al. (2005). The problems are scalable in both input dimension $d$ and in the number of objectives $M$. We set the input dimension to $d = 2M$ for each problem. The shape of the Pareto front of each of these problems can be viewed in the Pymoo documentation (`https://pymoo.org/problems/many/dtlz.html`).

Table 4: Specification of reference points used for all benchmark problems.

| Problem | Reference Point Botorch | Reference Point Pymoo |
|---|---|---|
| DTLZ1 | $[-400]^M$ | $[400]^M$ |
| DTLZ2 | $[-1.1]^M$ | $[1]^M$ |
| DTLZ3 | $[-10000]^M$ | $[10000]^M$ |
| DTLZ5 | $[-10]^M$ | $[10]^M$ |
| DTLZ7 | $[-15]^M$ | $[15]^M$ |
| Vehicle Safety | $[-1864.72, -11.82, 0.20]$ | Not performed |
| Pennicilin | Not performed | $[-1.85, 86.93, 514.7]$ |
| Car Side Impact | $[-45.4872, -4.5114, -13.339, -10.3942]$ | Not performed |
| Water Resource Planning | Not performed | $[84793, 1482, 3110300,$ $17141000, 381410, 103170]$ |
| Conceptual Marine Design | Not performed | $[-493.7, 17126, 5113.6, 14.35904]$ |

**Vehicle Crash Safety (RE3-5-4):** The vehicle crash safety problem is described in Tanabe & Ishibuchi (2020) and is a 3-objective problem with $d = 5$ parameters describing the size of different components of the vehicle's frame. The goal is to minimize mass, toe-box intrusion (i.e. vehicle damage), and acceleration in a frontal collision. We use the BoTorch implementation.

**Penicillin:** The Penicillin problem is described in Liang & Lai (2021) and is a 3-objective problem with $d = 7$ parameters. The goal is to maximize the penicillin yield while minimizing fermentatation time and CO2 byproduct. The input parameters are the culture volume, biomass concentration, temperature, glucose concentration, substrate feed rate, substrate feed concentration, and $H^+$ concentration. We used the default BoTorch implementation of this problem.

**Car side impact (RE-4-7-1):** The car side impact problem is described in Tanabe & Ishibuchi (2020) and is a 4-objective problem with $d = 7$ parameters. The goal is to minimize the car weight, the force experienced by a passenger, and the average velocity of the V-pillar responsible for withstanding

the impact load. The fourth objective is the sum of 10 constraint violations. We use the default implementation of this problem in BoTorch and Pymoo. Note this is a 3 objective problem in Pymoo.

**Water Resource Planning (RE-6-3-1):**   The Water Resource Planning problem is described in Tanabe & Ishibuchi (2020) and is a 6-objective problem with $d = 3$ parameters. The goal of the problem is to minimize the drainage network cost, the storage facility cost, the treatment facility cost, the expected flood damage cost, and the expected economic loss due to flood, of a water resource planning. The sixth objective is sum of seven contraint violations. The reference point was set using the infer_reference_point heuristic in BoTorch on the Pareto frontier over a large discrete set of random designs.

**Conceptual Marine Design (RE4-6-2:**   The Conceptual Marine Design problem is described in Tanabe & Ishibuchi (2020) and is a 4-objective problem with $d = 6$ parameters. The goal is to minimize the transportation cost, the light ship weights and the annual cargo transport capacity. The fourth objecdtive is the sum of nine constraint violations. The reference point was set using the infer_reference_point heuristic in BoTorch on the Pareto frontier over a large discrete set of random designs.

# D ADDITIONAL RESULTS

## D.1 TIMING RESULTS

All computations shown in Fig. 2 were performed on an Intel(R) Xeon(R) CPU E5-2640 CPU v4. and in the case of the GPU calculations on a NVIDIA TITAN X.

We now show the comparison of the Mean Absolute Percentage Error (MAPE) versus the computation for DeepHV and the approximate MC method (See Sec. 5.1 for details) for the cases of $M = 3$, $M = 5$, $M = 8$ and $M = 10$, in Fig. 5. It is shown that in all cases, DeepHV obtains a significantly lower MAPE at similar runtimes. The approximate MC method requires much more computation time to obtain similar a MAPE, which significantly increases at higher objective case $M$.

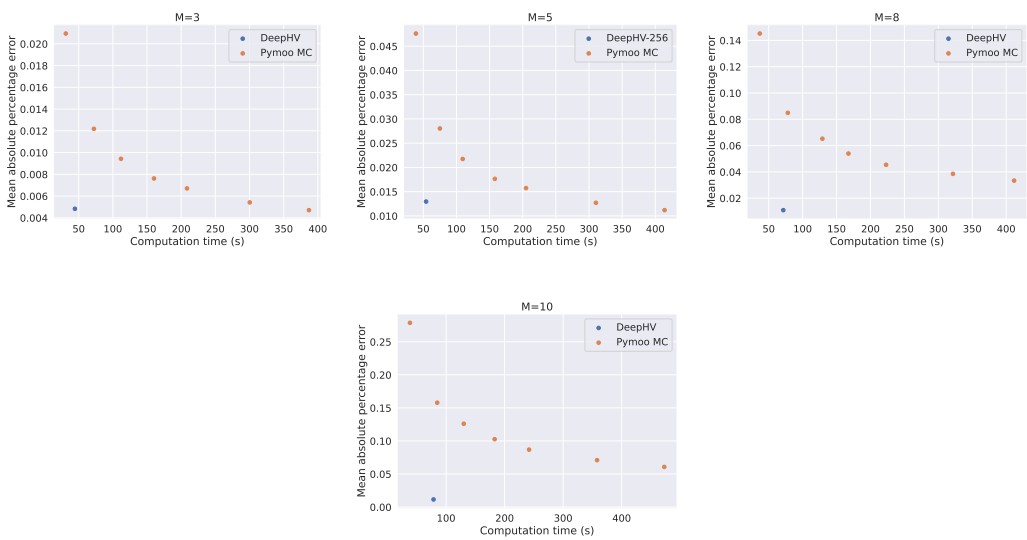

Figure 5: Comparison of computation time in seconds of the computation of 10000 solution sets randomly drawn for the objective cases $M = 3$, $M = 5$, $M = 8$ and $M = 10$. All computations performed on an Intel(R) Xeon(R) CPU E5-2640 CPU v4.

### D.2 MULTI-OBJECTIVE EVOLUTIONARY ALGORITHMS

#### D.2.1 WALL-TIME COMPARISON

Figure 6 provides a comparison of the wall time for the experiments shown in Fig. 3. We observe that SMS-EMOA, using the exact hypervolume implementation in Pytorch, is slowest, as is also supported by the results shwon in Sec. 5.1, its use become prohibitively slow for $M > 5$. NSGA-II and NSGA-III are fastest. The MC approximation (used in Water Resource Planning), is, using the current implementation, faster than the DeepHV models. It should be noted that the current implementation of the DeepHV models into Pymoo is not as efficient as it can be. Firstly, at the moment, the hypervolume of the front and all its sub-fronts now need to be predicted in an iterative fashion, which is not ideal, especially on CPU. DeepHV could also be trained to output the hypervolume contributions of all solutions in the front in one go, as in (Shang et al., 2022b), which would likely give an order of magnitude speedup. This should be studies in future research. The UCB-DeepHV-256-all model is considerably slower than UCB-DeepHV-128 and UCB-DeepHV-256 as it requires padding of the input to shape $100 \times 10$ and thus requires additional computations. CPUs are not particularly suitable for these computations, and hence these wall times could further benefit from the use of GPUs.

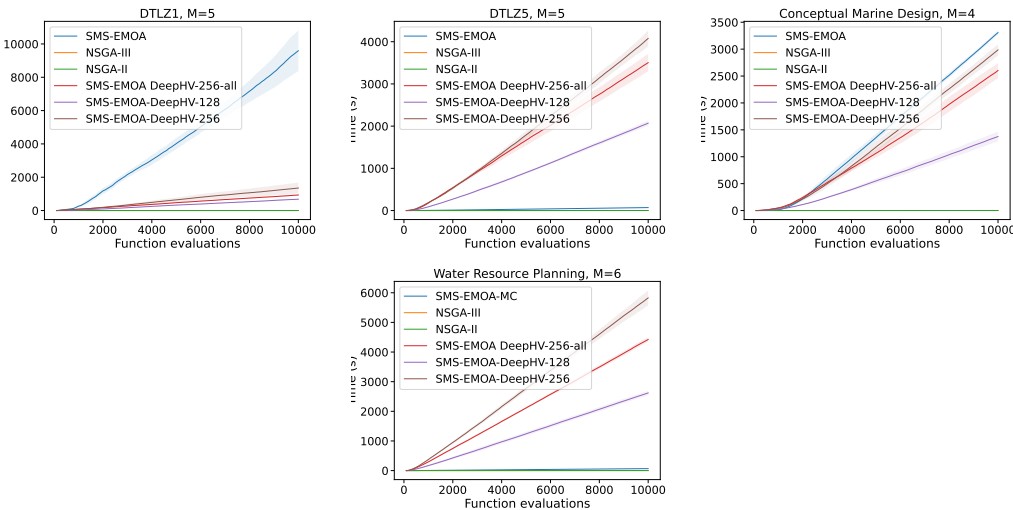

Figure 6: Cumulative wall time of experiments shown in Fig. 3. All computations were performed on an Intel Xeon Silver 4110 CPU. The mean and 2 standard errors over 5 trials are reported.

#### D.2.2 ADDITIONAL EXPERIMENTS

To study the performance of DeepHV in the context of multi-objective evolutionary algorithms on a broader range of problems and Pareto front shape, we ran additional experiments on a range of problems shown in Fig: 7. If $M > 5$ we use the MC approximation in SMS-EMOA. When comparing DTLZ1 ($M = 3$) with DTLZ1 ($M = 5$, in main text) and DTLZ1 ($M = 8$), it can be seen that in higher dimensions, the performance of NSGA-II drops significantly. The performance of the DeepHV models also deteriorates slightly, but they are still able to find a relatively good approximation of the Pareto front, albeit not as diverse as for NSGA-III and SMS-EMOA. Of course, the 8 dimensional objective space is much larger than the 3 objective space, and perhaps more training data is required to have better performance. This behavior is also observed for DTLZ5 but to a lesser extent. DTLZ2 has a Pareto front that lies on the first octant of the unit hyper sphere, i.e. it has a very idealistic concave shape. The DeepHV models find some points on the unit sphere, but fails to have a nicely spread population over the sphere compared to SMS-EMOA and NSGA-III. In the case $M = 5$, it still outperforms NSGA-II. DTLZ3 has the same Pareto front shape as DTLZ2 but is parametrized differently, in addition we optimized this problem with a differently set reference point $[10000^M]$. This effectively changes the shape of the Pareto front, and now all models perform similarly. DTLZ7 ($M = 3$) has four disjoint Pareto-optimal regions, which generally is a good test

to see if subpopulation is maintained in each region, which is done similarly well for all methods. DTLZ3

Figure 7: Optimization performance in terms of hypervolume (higher is better) versus function evaluations. We report the means and 2 standard errors across 5 trials.

## D.3 MULTI-OBJECTIVE BAYESIAN OPTIMIZATION

### D.3.1 WALL-TIME COMPARISON

Figure 8 provides a comparison of the wall time for the experiments shown in Fig. 4. For the Car Side Impact and DTLZ2 ($M = 4$) problems, we show plots with and without UCB-HV for clearer comparison. In addition, we show timings for DTLZ2 ($M = 6$) to show scaling of qEHVI. We observe that UCB-HV, using the exact hypervolume implementation in BoTorch, is slowest, as is also supported by the results shown in Sec. 5.1. Its use becomes prohibitively slow for $M > 4$. qParego generally is fastest and runtime scales well with increasing objectives. qEHVI, which has a highly efficient parallel implementation, generally is in the same ballpark with UCB-DeepHV-128 and UCB-DeepHV-256 for problems up to $M = 5$. However, above $M = 5$, qEHVI is considerably slower than the other methods. The UCB-DeepHV-256-all model is considerably slower than UCB-DeepHV-128 and UCB-DeepHV-256 as it requires padding of the input to shape $100 \times 10$ and thus

requires additional computations. CPUs are not particularly suitable for these computations, and hence these wall times could further benefit from the use of GPUs.

Figure 8: Cumulative wall time of experiments shown in Fig. 4. All computations were performed on an Intel Xeon Silver 4110 CPU. The mean and 2 standard errors over 5 trials are reported.

### D.3.2 ADDITIONAL EXPERIMENTS

In order to further evaluate the performance of DeepHV in the context of multi-objective Bayesian optimization, we also report results for the DTLZ1, DTLZ2, DTLZ3, DTLZ4 and DTLZ5 synthetic test problems for the objective cases $M = 3, 4$ and $5$ if not otherwise shown in the main text. In addition, we show results on the real-world Penicillin and Vehicle Safety problems. In the case of UCB-HV, we do not show results for the case $M = 5$ and the DTLZ4 problems. These results are shown in Fig. 10. The other resuts are shown in Fig. 9. All experiments were performed 5 times, and we report the mean and 2 standard errors. In general, DeepHV is competitive with UCB-HV, qParego and qEHVI, although qEHVI clearly performs best on DTLZ2, DTLZ4 and Vehicle Safety.

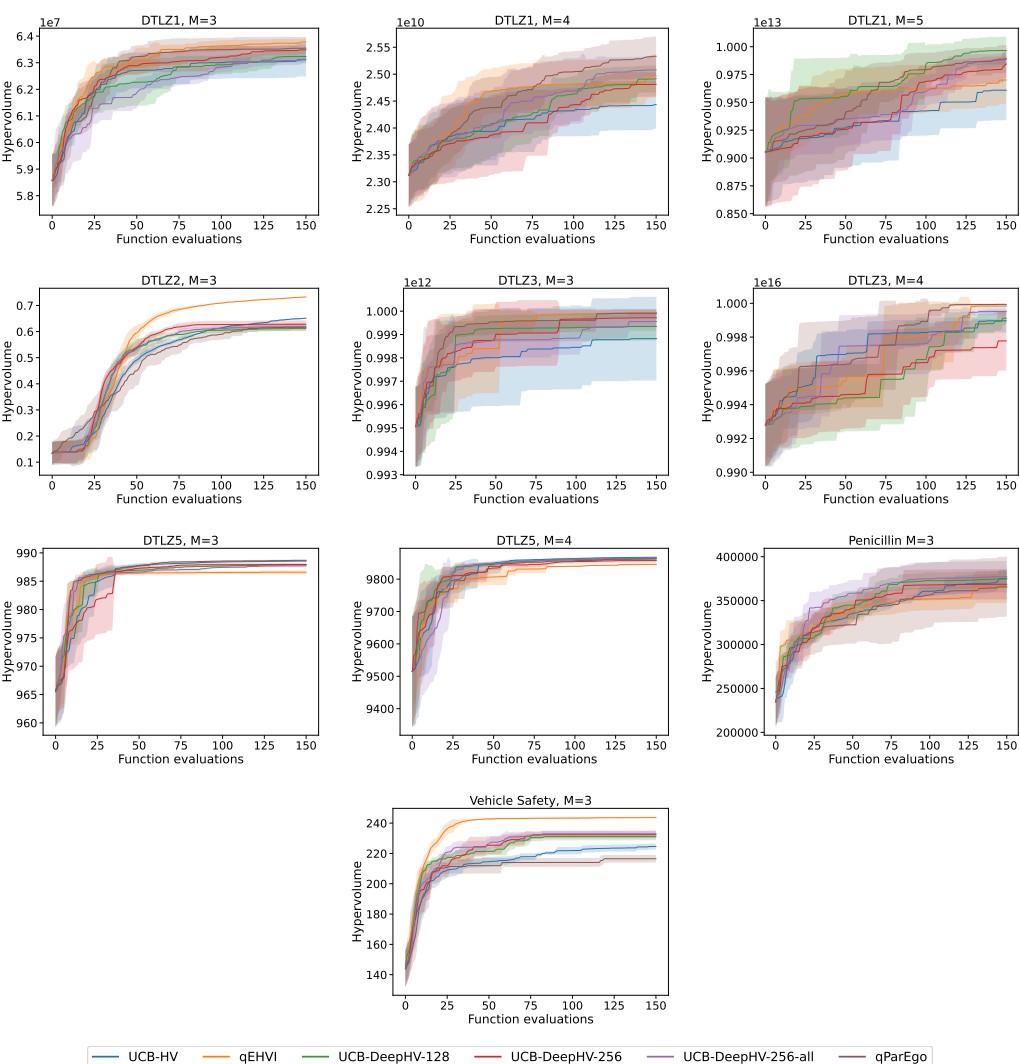

Figure 9: Sequential optimization performance in terms of hypervolume (higher is better) versus function evaluation.

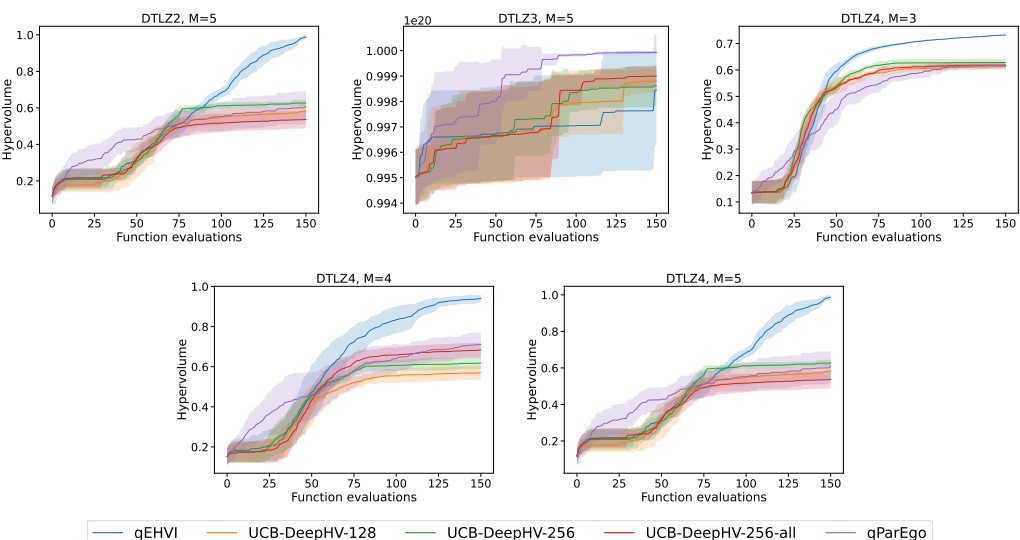

Figure 10: Sequential optimization performance in terms of hypervolume (higher is better) versus function evaluation.

