# OpenReview forum: "Multi-objective optimization via equivariant deep hypervolume approximation"
_ICLR.cc/2023/Conference — ICLR 2023 poster_

### Official Review · Reviewer_9RfZ · 2022-10-23

**Confidence:** 4
**Correctness:** 4
**Technical Novelty And Significance:** 3
**Empirical Novelty And Significance:** 3
**Recommendation:** 6

**Clarity, Quality, Novelty And Reproducibility:**

**Clarity**

The paper is well-written with good clarity explaining all relevant pieces.

**Quality**

The method description and experiments generally show good quality. I think answers to some of the above questions could further strengthen the quality.

**Novelty**

The paper proposes a novel method for a relevant problem in multi-objective optimization.

**Reproducibility**

The authors provide clear description of the method and a reproducibility statement. Assuming the code gets released, as indicated in the reproducibility statement, other should be able to reproduce the results.

**Strength And Weaknesses:**

**Strengths**

* The paper proposes a novel methods in a relevant challenge in multi-objective optimization and show promising results for making hypervolume calculations computationally feasible in large-scale search with high numbers of objectives and data points.
* The paper proposes a new G-equivariant layer based on first principles and provides results supporting their method compared to current methods (HVNet).
* The paper shows experiments with DeepHV used in modern multi-objective optimization methods.

**Weaknesses**

* The paper lacks an ablation on the equivariant layers to clearly show what benefits they provide. It might be possible that the HVNet performance provides that ablation in an indirect sense, so it would be good for the authors to clarify this.
* The generalization claim of the authors related to DeepHV does not appear to well-tested. Right now it seems that DeepHV trains on data for the same problems that it is applied on and one could argue that it somewhat generalizes to the distribution of hypervolume in those design problems. It would be nice to explore this notion further as greater generalization would provide a stronger case for DeepHV.
* SMS-EMOA with Monte-Carlo estimation still outperforms DeepHV in terms of task performance. It might be interesting to see if the compute cost of SMS-EMOA is higher compared to the DeepHV methods. That might showcase an interesting trade-off between the methods worth exploring.

**Additional Questions**

* Could you provide an intuitive explanation for the mathematics in Section 3.1. This would complement the current section and proof in the appendix (which would be good to reference in the main text).
* What is the importance of generating data only non-dominated solutions? Wouldn't hypervolume calculations be equally valid with dominated solutions?
* It would nice to see to potentially have a measure of compute cost on the x-axis instead of function evaluations. Assuming you can do that analysis, does it show that DeepHV consistently saves on compute? If so, in what cases?
* Any intuition why training with multiple dimensions of objectives does not help the 3-objective case in Table 2?
* What is the primary source of noise in the multi-objective BO settings?

**Summary Of The Paper:**

The paper proposes a new method leveraging equivariant deep learning to approximate hypervolume calculation in multi-objective optimization (DeepHV). The authors first introduce important concepts in multi-objective optimization, such as problem formulation and the Pareto optimality. The authors then describe current hypervolume calculation methods (WFG, HBDA, FPRAS, HV-Net) and their scaling limitations in terms of number of objectives (M) and number of data points (N) and how hypervolume are used or avoided in current multi-objective optimization methods, such as multi-objective Bayesian optimization and multi-objective evolutionary algorithms. Following the introduction, the authors formally define the hypervolume calculation and it's important symmetric properties (permutation invariance and G-equivariance) which are then used to propose a G-equivariant variation of a fully-connected neural network layer that make up the building block of DeepHV. The Deep HV network used in the experiments includes five layers of G-equivariant layers proposed by the authors.

For their experiments, the authors generate $\sim$ 1M training data points by sampling solutions for different dimension of objectives (3-10) that are then used to train the DeepHV approximator. The first set of experiments presented show that DeepHV achieves a lower MAPE compared to HVNet and generally performs better with a greater set of parameters. Subsequent experiments show runtime benefits of DeepHV compared to open-source hypervolume calculations in Pymoo and Botorch. The final set of experiments of experiments involve using DeepHV in active multi-objective EA and BO settings. In the multi-objective EA settings, DeepHV performs better than NSGA2 but fails to beat SMS-EMOA with a traditional Monte-Carlo calculator across various multi-objective optimization benchmarks. For multi-objective BO, the results are mixed with traditional methods sometimes outperforming DeepHV methods and vice versa.

**Summary Of The Review:**

Overall, the paper is well-written and proposes a novel in a relevant problem settings which provide good reasons to argue for acceptance. Nevertheless, I think the paper can still be improved by addressing current weaknesses, such as providing a more detailed description related to generalization and compute costs of DeepHV compared to other methods.

---

> ### Comment · Reviewer_9RfZ · 2022-11-16
> **Author Response**
>
> I appreciate the authors' thorough response to the questions and comments. At this moment, I will keep my current score and will re-evaluate once the draft has been updated with the additional results, such as the equivariant layer ablation.

---

> > ### Author Response · Authors · 2022-11-19
> > **Response**
> >
> > Thanks for your comment, we have now updated the draft.

---

> > > ### Author Response · Authors · 2022-11-30
> > > **Response**
> > >
> > > We would like to check in to see if the updated manuscript has been received well and if there are any unclarities regarding the added ablation study.

---

> > > > ### Comment · Reviewer_9RfZ · 2022-12-06
> > > > **Update**
> > > >
> > > > Thank you for the updated draft and additional details. After reviewing the changes, I maintain my current score.

---

### Official Review · Reviewer_DvS5 · 2022-10-24

**Confidence:** 3
**Clarity, Quality, Novelty And Reproducibility:** 1.	The advantages of the proposed met…
**Correctness:** 3
**Technical Novelty And Significance:** 3
**Empirical Novelty And Significance:** 2
**Recommendation:** 6

**Strength And Weaknesses:**

Strengths
1.	A new deep neural network-based approximation with permutation and scale equivariance is proposed for the hypervolume function.
2.	The organization is clear and easy to follow.

Weaknesses
1.	The advantages of the permutation and scale equivariance in the hypervolume approximation need more discussion.
2.	The effectiveness of the proposed method needs more evaluation. First, when comparing the effectiveness of training a model on each objective case separately with that on all objective cases simultaneously, how is the experiment set to ensure fairness? Second, it is reported that the boost in performance of DeepHV compared to HV-Net is likely due to the incorporation of the symmetry properties of DeepHV. How? Third, the results shown in Section 5.2.1 do not appear to be consistent with that in Section 4.2. For example, SMS-EMOA that uses exact computation mostly achieves the best results, yet the approximation models (e.g., DeepHV-256-all) that are shown to achieve higher accuracy in Section 4 rarely show better results in Section 5. This raises the question of whether the metrics used in Section 4 are appropriate. Fourth, the experiments conducted in Section 5 should include more state-of-the-art (approximated) hypervolume-based methods.


**Summary Of The Paper:**

This paper proposes a new method to approximate the hypervolume function with a deep neural network The network is built by using specialized layers to incorporate some symmetry properties of the hypervolume function, such as permutation and scale equivariance.

**Summary Of The Review:**

This paper proposes a new deep neural network-based hypervolume approximation method that is equipped with some symmetry properties. More discussions and more evaluations are needed to verify the advantages of the proposed method.

---

### Official Review · Reviewer_cbGc · 2022-10-24

**Confidence:** 3
**Correctness:** 4
**Technical Novelty And Significance:** 3
**Empirical Novelty And Significance:** 3
**Recommendation:** 6

**Clarity, Quality, Novelty And Reproducibility:**

        The paper is clearly written and difficult concepts are clearly explained.


**Strength And Weaknesses:**

Strengths:

        - Well written paper.

        - Original architecture proposed to compute the hyper-volume.

        - Extensive experiments evaluating performance and computational time.

Weaknesses:

        - It seems only synthetic problems are considered in the experiments.

        - The authors claim in the discussion that their approach outperforms MC approximations in terms of performance. However, the experiments only seem to indicate that the proposed approach is faster. MC approximations seem to perform better.

        - No error curves are given in Table 1 and 2. However, since the test set is large. E.g. 100k points it may not be a problem.



**Summary Of The Paper:**

        This paper proposes a deep neural network to approximate the computation of the hyper-volume, which becomes expensive for a large number of objectives or points. The network has a special architecture to take into account the particularities of the hyper-volume computations such as invariances to permutations. The model is compared with state-of-the-art approaches, showing lower prediction error. It also compares favorably to the exact computation and other approximations in the context of evolutionary algorithms and Bayesian optimization. The computation speed is better also for a larger number of objectives.


**Summary Of The Review:**

Overall I think that this is an interesting paper that proposes a new method that will receive the attention of the community. The paper is well written and the experimental section is strong, with experiments that clearly show the benefits of the propose approach. My only concern is that only synthetic experiments seem to have been carried out.

Minor:

        There are other methods that for multi-objective BO that do not use the hyper-volume directly, but that need to
        solve a multi-objective problem. For example, see:

        Predictive entropy search for multi-objective Bayesian optimization
        D Hernández-Lobato, J Hernandez-Lobato, A Shah, R Adams
        International conference on machine learning, 1492-1501

---

### Official Review · Reviewer_favf · 2022-10-24

**Confidence:** 4
**Correctness:** 3
**Technical Novelty And Significance:** 3
**Empirical Novelty And Significance:** 3
**Recommendation:** 6

**Clarity, Quality, Novelty And Reproducibility:**

**Clarity:** This paper is generally well-organized and easy to follow, but many important details and discussions are missing as discussed in the weaknesses above.

**Quality:** The overall quality is good, but the concerns listed in the weaknesses section should be carefully addressed.

**Novelty:** The original idea of model-based hypervolume approximation, problem formulation (e.g., problem definition and training data generation), and model framework are already proposed in the previous work HV-Net [1]. Therefore, the novelty of this work is somehow limited.

**Reproducibility:** It seems the proposed model can be easily reproduced, but many additional experiments are required to truly analyze the pros and cons of the proposed model.

**Strength And Weaknesses:**

**Strengths:**

+ This paper is generally well-organized and easy to follow.

+ Efficiently calculating the (approximate) hypervolume, especially for problems with many objectives (e.g., >= 5), is an important while challenging problem for multi-objective optimization. This work is a timely contribution to an important research topic for multi-objective optimization.

+ The proposed model can achieve good performance for hypervolume approximation, and can be easily used in the current EA/BO methods for multi-objective optimization.

**Weaknesses:**

**1. Novelty and Contribution**

The original idea of model-based hypervolume approximation, problem formulation (e.g., problem definition and training data generation), and model framework are already proposed in the previous work HV-Net [1]. This work's main contribution is the new equivariant layer for better approximation performance, which is a natural extension from the equivariant layer proposed in [2]. Although it is glad to see the proposed model can achieve better performance (as listed in the strengths), the contribution is not very novel given the previous works.

In addition, the close relation to the previous work HV-Net should be clearly discussed and emphasized early in this paper. I did believe this work has proposed the idea of model-based hypervolume approximation until the end of page 2. The claimed contribution in the abstract, "To overcome these restrictions, we propose to
approximate the hypervolume function with a deep neural network, which we call DeepHV", is also a bit misleading.

**2. Unclear Comparison with HV-Net**

Since the main contribution is the improvement over HV-Net, their difference should be carefully discussed in detail, which is missing in the current paper.

*a) Symmetry Properties*

For example, this work claims DeepHV can "leverage additional symmetry properties of hypervolume compared to HV-Net". It seems that both DeepHV and HV-Net have the permutation invariant property, while DeepHV additionally has the scale-equivariance of the objective values. Do the additional symmetry properties mean the scale-equivariance to objective values? Why and how this additional property can lead to better Hypervolume approximation?

In addition, it could be beneficial to have an ablation study and detailed analysis for the layer design (21) to clearly show the effect of a) no permutation invariance, no scale equivariance; b) only permutation invariance, but no scale equivariance; c) no permutation invariance, but only scale equivariance; d) both permutation invarince, and scale equivariance for hypervolume approximation. Is HV-Net corresponding to the case of b), and the proposed DeepHV is the case of d)?

Could a simple normalization of the objective values (e.g., all to [0,1]) significantly improve the approximation performance of HV-Net? It seems that the related order of HV could be more important than the approximation accuracy for optimization.

*b) Comparision Setting*

Some comparison settings with HV-Net might need further clarification.

The DeepHV model is both trained and tested with the MAPE loss, while the HV-Net is trained with the log MSE loss and tested on the MAPE loss. Will it lead to unfair comparison?

For DeepHV model, the training data is split into train-validation-test sets, so the hyperparameters and settings can be carefully fine-tuned on the validation set. It seems that the HV-Net is directly trained on the whole dataset without fine-tune. Will it lead to unfair comparison?

**3. Test Problems and Reported Results**

It is glad to see DeepHV can have promising performance with both EA and BO approaches to solve multi-objective optimization problems. However, the current reported results are not solid enough to truly show DeepHV's advantages.

This work uses the DTLZ benchmarks for most experiments, and only one real-world problem is reported at the very end of the Appendix (e.g., page 18). The DTLZ benchmarks have a regular shape of PS, which might be very different from real-world problems. It could be much better to report the results on the real-world problems suite in [3] with 2-9 objectives, which is now widely used in both the EA and BO community.

Why only the results of DTLZ 1,2,5,7 are reported? How about DTLZ 3, 4, 6?

For all experiments, please report the value of the log hypervolume difference (e.g., as in the qEHVI paper [4]) rather than the hypervolume value. The found Pareto set with "similar" hypervolume value could perform very differently and have quite different values of log hypervolume difference.

**4. Experiments with EA**

For all EA experiments, only 1,000 function evaluation is far from the ideal setting. In the standard EA setting, all algorithms should be run with a large number of evaluations (e.g., 100,000) such that the found solutions can converge to the true Pareto front. Will the approximation nature of DeepHV make it hard to find the true Pareto solutions, and lead to poor performance in the standard EA setting?

For the baseline algorithm, it is well-known that the domination-based algorithm such as NSGA-II will perform poorly on problems with more than 3 objectives. Outperforming NSGA-II is not enough to show the advantage of DeepHV for many-objective optimization. NSGA-III [5] could be a much more suitable baseline for comparison (which is also available in Pymoo).

It is also interesting to know the wall-clock runtime of these algorithms for problems with different numbers of objectives.

**5. Experiments with BO**

For all BO experiments, the important results of UCB-Exact HV and UCB-MC HV are missing. Will UCB-DeepHV have similar performances with these baselines?

On the other hand, since UCB-DeepHV with different model sizes perform similarly to each other, will HV-Net also have similar BO performance?

It is interesting to know the performance of qEHVI with UCB acquisition function as in UCB-DeepHV.

**6. The Opening Story**

I personally enjoy reading the opening story in the introduction for multi-objective optimization. However, reading this paper while driving could be dangerous and indeed against the law in many countries. It is better to replace it with a harmless story.

**Other Comments**

1. In Figure 2, it is interesting to know the computation time on GPU for DeepHV and the exact hypervolume calculation in BoTorch.

2. If the goal of hypervolume approximation is to be used with EA or BO for optimization, it seems directly approximating the hypervolume contribution (HVC) could be a more reasonable choice. There is a concurrent work on HVC approximation (HVC-Net) [6], where a comparison is not needed, but a brief discussion could be beneficial.

[1] HV-Net: Hypervolume Approximation based on DeepSets. IEEE Transactions on Evolutionary Computation 2022.

[2] Deep models of interactions across sets. ICML 2018.

[3] An easy-to-use real-world multi-objective optimization problem suite. Applied Soft Computing 2020.

[4] Differentiable expected hypervolume improvement for parallel multi-objective Bayesian optimization. NeurIPS 2020.

[5] An evolutionary many-objective optimization algorithm using reference-point-based nondominated sorting approach, part I: solving problems with box constraints. IEEE Transactions on Evolutionary Computation 2014.

[6] HVC-Net: Deep Learning Based Hypervolume Contribution Approximation. PPSN 2022.


**Summary Of The Paper:**

This work proposes DeepHV, a deep neural network based model, to approximate hypervolume for multi-objective optimization. The idea of model-based hypervolume approximation has been proposed in the previous work (HVNet). This work's contribution is to design a more powerful equivariant layer to further exploit hypervolume's scale-equivariant and permutation invariant properties for better modeling. Experimental results show that DeepHV can better approximate the hypervolume, and can be well combined with EA and BO methods for multi-objective optimization.

**Summary Of The Review:**

This work is a timely contribution to an important research problem for multi-objective optimization. However, due to the concerns on novelty, unclear comparison with closely-related work, and experimental settings, I cannot vote to accept the current submission.

---

> ### Comment · Reviewer_favf · 2022-11-21
> **Thank You for the Thorough Response**
>
> Thank you for the thorough response and additional experiments. I have increased my score to 6.
>
> Here are a few follow-up comments that will not affect the score.
>
> **Ablation Study**
>
> It is interesting to know why the gap between DeepHV-no-scaling and DeepHV is significantly larger than the gap between MLP and MLP-with-scaling.
>
> **Log Hypervolume Difference**
>
> I agree with the author that the relative performance for different algorithms will not change from hypervolume to hypervolume difference. But the hypervolume difference is important to check each algorithm's absolute performance, such as whether the solution set actually converges to the true Pareto set. In addition, the hypervolume value itself could heavily depend on the selected reference point, so its individual value might not be very informative.
>
> For example, say algorithm A has HV_A = 99 and algorithm B has HV_B = 98, one might conclude that they have similar performance. But if the true Pareto set has HV = 99, we know the solution set found by A has converged to the Pareto set, but B does not. Furthermore, by selecting another reference point for HV calculation, we might get HV_A = 2 and HV_B = 1, and now they are not similar to each other anymore (against the original 99 v.s. 98). But the hypervolume difference for A to the true Pareto set could be still 0.
>
> **BO Experiments**
>
> From the results, UCB-HV (with exact value) and UCB-DeepHV with different sizes (and hence different approximation accuracies) have similar BO performance. It seems that better approximation accuracy will not necessarily lead to better BO performance. That is why I am curious about the HV-Net's performance.

---

### Author Response · Authors · 2022-11-15
**General Remarks**

## Word of thanks & outline
We like to thank all reviewers for their detailed reviews, comments, and suggestions. We feel the work has significantly improved due to suggestions such as the ablation study, which we had not thought of ourselves.

We first start out by addressing some remarks that multiple reviewers shared. Following that, we handle reviewer comments per reviewer individually. Not all comments have already been revised in the manuscript, and we will work hard to be able to upload the revised manuscript as early as possible before the deadline. We hope that by publishing these comments now, we provide you with enough time to assess our comments. **We will keep you up-to-date on manuscript changes when new content is added.**

## Ablation Study
Reviewers favf, DvS5 and 9RfZ if the proposed equivariant layers provide additional performance over not incorporating permutation invariance (over solutions and objectives) and scaling equivariance and if its performance gain can be quantified. Reviewers favf and 9Rfz propose doing an ablation study to investigate this. We indeed think this is a good idea and have done this. The ablation study consisted of the comparison of the proposed DeepHV architecture, the DeepHV architecture without scaling equivariance (thus only considering the permutation equivariances),  an MLP with scaling equivariance (thus only considering scaling equivariances), and an MLP without scaling equivariance (thus considering no symmetries). We performed this analysis for the cases M=3, M=5, M=8, and M=10. We used a similar number of parameters for each model. In addition, we notice that without scaling equivariance, the mean absolute percentage loss is unstable, therefore we also explore the use of other loss functions for a fair comparison. We describe this in Section 4.3 and will update the manuscript asap.

## Novelty & Contributions
There were some remarks that this work is not very novel and has much overlap with the HV-Net paper. However, we would like to point out some differences which were potentially overlooked.
- We consider 2 extra symmetries:
So there are three symmetry properties of the hypervolume that we exploit, scale-equivariance of the objective values, permutation invariance over the solutions in the solution set, and permutation over the objectives. We would like to stress that HV-Net only considers the permutation invariant property over the solutions in the solution set (using DeepSets), and not the permutation invariance over the objectives themselves. This is stated on page 2, as part of the introduction. Therefore, the scale-equivariance is not the only addition, we also leverage the permutation invariance over objectives.
- We now perform an ablation study to point out the importance of incorporating these symmetries for both the stability of the training process, but also for the performance of the model
- In the HV-net paper, no experiments were performed to test its performance for multi-objective optimization tasks, we are the first to actually do this. Both in the setting of BO and EAs.

## More Benchmarks & longer runtimes
Reviewers favf and cbGc asked for more real-world benchmarks, we, therefore, have performed additional experiments on these. Reviewer favf kindly pointed us to another problem suite [3]. From this suite, we now also incorporated experiments on the RE4-7-1 problem (Car side impact design, M=4) in the BO experiments. In addition, for the BO experiments, we have now also included the Pennicilin problem (M=3) [7]. In addition, we have implemented the RE6-3-1 (Water Resource Planning, M=6) and RE4-6-2 (Conceptual Marine Design, M=4)) from this suite [3]. We will add the results of these experiments to the manuscript as soon as possible.

In addition, upon request of reviewer favf we are running all experiments performed in the multi-objective EA setting with more function evaluations.

[3] An easy-to-use real-world multi-objective optimization problem suite. Applied Soft Computing 2020.
[7] Scalable Bayesian Optimization Accelarates Process optimization of Pennicilin Production: NeurIPS 2021

## Runtime comparison
Reviewers favf and 9RfZ pointed out that this work would benefit from the wall-clock runtime of the optimization algorithms for problems with different numbers of objectives. We initially refrained from this as there are other factors than just the hypervolume computation that influence the runtime. For instance, in the BO setting, fitting the gaussian process can have variable timings depending on the optimization “route”. In addition, optimizing the acquisition function may need varying iterations and thus a variable amount of function evaluations (HV computations in qEHVI and UCB-HV). Therefore we initially decided to provide a more pure time comparison as in Figure 2. We do agree that adding wall-clock runtime does make this work stronger and we will thus provide wall-clock time results.

---

### Decision · Program_Chairs · 2023-01-20

**Decision:**

Accept: poster

**Justification For Why Not Higher Score:**

- Lack of novelty

**Justification For Why Not Lower Score:**

- Paper presents a new method that performs well in an important problem, as demonstrated convincingly by experimental evaluations

**Metareview: Summary, Strengths And Weaknesses:**

This paper considers multi-objective optimization. A common approach to this problem is to quantify the value of a set of proposed Pareto-optimal points by their hypervolume. A challenge in doing this is that the hypervolume can take a significant amount of time to compute, particularly in high-dimensional problems and when this calculation is repeated many times as part of a larger algorithm (e.g., an evolutionary algorithm, EA, or Bayesian optimization, BO).

This paper prosposes a deep architecture to predict the hypervolume, called DeepHV. It then uses this prediction within EA and BO algorithms. These methods are used to demonstrate good empirical performance on synthetic and real-world examples.

There is another paper written around the same time period that proposed a method called HV-Net (Shang et al. 2022a) that is similar. In comparison to this paper, their architecture considers two extra symmetries of hypervolumes and demonstrates experimentally that hypervolume prediction is useful for multi-objective optimization. These symmetries are shown to be important in an ablation study.


Strengths
- The reviewing team felt that the paper was well-written and easy to follow
- The paper demonstrated their method in a full optimization loop and demonstrated that fast hypervolume estimation was useful for optimization, in contrast with past HVNet work
- Good empirical results

Weaknesses
- Lack of novelty. The idea of predicting hypervolume with a deep architecture is not new (see HV-Net discussion above). Methods for incorporating set symmetries are taken from Deep Sets (Zaheer et al. 2017). The reviewing team felt that it is was critical for the authors to openly and honestly discuss novelty early in the paper.
- There are many previously proposed multi-objective algorithms, but few were considered in experiments
- The direct evaluations of predictive accuracy are in-sample, though the evaluation on multi-objective optimization is, in a sense, out of sample

**Note From Pc:**

if the above contains the word "oral" or "spotlight" please see: "oral" presentation means -> notable-top-5% and "spotlight" means -> notable-top-25%. As stated in our emails, we are disassociating presentation type from AC recommendations

**Summary Of Ac-Reviewer Meeting:**

Much of the meeting discussed the importance of novelty in making a decision about this paper. As discussed in the meta-review, this is in question. The AC remains a bit uncomfortable with the lack of novelty, but the reviewers felt that the paper's novelty was sufficient for publication.

We also discussed experimental evaluation, deciding that while there are some issues with the evaluation it is overall reasonably good.